# The Cancer Stem Cell Niche in Ovarian Cancer and Its Impact on Immune Surveillance

**DOI:** 10.3390/ijms22084091

**Published:** 2021-04-15

**Authors:** Srishti Jain, Stephanie L. Annett, Maria P. Morgan, Tracy Robson

**Affiliations:** School of Pharmacy and Biomolecular Science, RCSI University of Medicine and Health Sciences, 123 St Stephen’s Green, D02 YN77 Dublin, Ireland; SrishtiJain@rcsi.ie (S.J.); stephanieannett@rcsi.com (S.L.A.); mmorgan@rcsi.ie (M.P.M.)

**Keywords:** ovarian cancer, cancer stem cells, immune surveillance, tumour microenvironment

## Abstract

Ovarian cancer is an aggressive gynaecological cancer with extremely poor prognosis, due to late diagnosis as well as the development of chemoresistance after first-line therapy. Research advances have found stem-like cells present in ovarian tumours, which exist in a dynamic niche and persist through therapy. The stem cell niche interacts extensively with the immune and non-immune components of the tumour microenvironment. Significant pathways associated with the cancer stem cell niche have been identified which interfere with the immune component of the tumour microenvironment, leading to immune surveillance evasion, dysfunction and suppression. This review aims to summarise current evidence-based knowledge on the cancer stem cell niche within the ovarian cancer tumour microenvironment and its effect on immune surveillance. Furthermore, the review seeks to understand the clinical consequences of this dynamic interaction by highlighting current therapies which target these processes.

## 1. Introduction

In 1992, ovarian cancer was termed ‘the most lethal gynaecologic malignancy’ [1], with the overall five-year survival rate reported at 30%. Although the past three decades have seen a significant improvement in diagnostic advances, therapeutic strategies and overall care in ovarian cancer, prognosis continues to remain poor. The current five-year survival rate of 48.6% is the lowest among all gynaecological cancers [2], meriting the dismal title of ovarian cancer being the deadliest gynaecological cancer. Over 90% of all ovarian cancers are of epithelial origin and can be broadly divided further into Type I (including low- grade serous, endometrioid, clear-cell or mucinous carcinomas) and Type II (including high-grade serous or undifferentiated carcinomas).

Population-based cancer incidence and mortality data is compiled by various organisations across the world. For Europe, the European Cancer Information System estimates an age standardised incidence rate of ovarian cancer at 16.1 per 100,000 and an associated mortality rate of 10.4 per 100,000 (Figure 1) [3]. This high mortality-to-incidence ratio is attributable to a combination of late detection and resistance to therapy. The improbability of early diagnosis is a direct consequence of the lack of specific symptoms during the early stages of the disease, as well as the absence of reliable screening strategies. Owing to the success of cervical and breast cancer screening, as well as the rather modest increase in survival from improved treatment, there have been fervent efforts to boost ovarian cancer survival via screening using CA125, an epitope of MUC16, a large glycoprotein marker. However, the accuracy of this biomarker is still questionable, although more effective screening strategies with CA125 are being developed [4]. As outlined before, therapeutic advances have led to only a small increase in ovarian cancer survival rate over the years. Standard treatment for ovarian cancer is cytoreductive surgery along with combination taxane­–platinum-based chemotherapy. More recently, the two most promising novel therapeutic approaches are using monoclonal antibodies such as bevacizumab, targeting tumour microenvironmental pathways such as angiogenesis, and inhibitors of the poly (ADP-ribose) polymerase (PARP) enzyme which is involved in critical cellular functions such as DNA repair. Both have been approved by the FDA and show promising outcomes as combinatorial and maintenance drugs in ovarian cancer [5].

Although first-line therapy has an initial remission rate of 70–80%, the majority of patients relapse, develop chemoresistance and proceed to respond only very modestly to second-line chemotherapy. The high recurrence rate and chemoresistance associated with ovarian cancer is thought to be due to intra-tumoral heterogeneity, microenvironmental interactions as well as the presence of dynamic cancer stem cell sub-populations. There are three main models proposed to explain the heterogeneity of intra-tumoral cell populations. The two conventional models are the clonal evolution or stochastic model and the stem cell or hierarchical model. It is now understood that the two ideas are not mutually exclusive, and a third model termed the plasticity model conceptualises a more dynamic, flexible understanding of the tumoral niche (Figure 2). Stem cell-like subpopulations existing in the tumoral hemisphere in solid tumours such as ovarian cancer have been found to dynamically interact with the immediate cellular microenvironment so as to induce tumorigenesis, survival and metastases as well as self-renewal leading to an intrinsically generated and maintained tumour niche capable of immunosuppression and therapeutic evasion. Hence, it is vital to study these interactions and devise methods that effectively target these stem cell niches to make substantial strides in the therapeutic targeting and management of aggressive ovarian tumours. This review aims to summarize the current understanding of the ovarian cancer stem cell niche and its interactions with the host immune system and to highlight implications for the development of novel ovarian cancer therapies.

## 2. Ovarian Cancer Stem Cells (OCSCs): Signaling Pathways and Markers

Like many solid tumours, ovarian cancer has been shown to reflect significant tumoral phenotypic diversity [6]. Key evidence suggests that the high relapse rate inevitably seen in ovarian cancer is linked to chemoresistant stem cell-like subpopulations which persist through therapy and have tumorigenic properties [7]. In 2013, Virant-Klun et al., first discovered very small embryonic-like stem cells identifying stage-specific embryonic antigen-4- (SSEA-4; a marker of human embryonic stem cells) positive cells from cultures of human ovarian cancers and validated their discovery in women with borderline ovarian cancer (a less aggressive form of epithelial ovarian cancer) versus healthy women. The cells from the test group were proliferative and formed tumour-like structures in vitro as well as in vivo [8,9].

### 2.1. Signaling Pathways

A number of oncogenic signaling pathways have been found to generate and maintain OCSCs, as summarised in Figure 3. Specific inhibition of these pathways has shown promising results in decreasing stemness in ovarian cancer cell lines as well as in animal models and will be discussed later in the review.

Notch3 has been found to be overexpressed in high-grade serous ovarian cancer (HGSOC) [20]. In ovarian cancer cell lines, Notch3 overexpression causes upregulation of pathways associated with stem cell generation. Treatment of ovarian cancer cells with notch pathway inhibitors was found to deplete stem cells and when administered in combination with cisplatin, it eliminated the stem cell population as well as the tumour cells [21]. The Wnt pathway has been implicated in the ovarian cancer stem cell niche. Specific G-protein-coupled receptors have been associated with Wnt pathway regulation of stem cells in the ovary [22]. Downstream β-catenin activation leads to upregulation of ABC transporters, which have been linked to the development of taxane–platinum therapy resistance [23]. PTCH1 and Gli1 transcription factors associated with the Hedgehog pathway have also been found to be overexpressed in ovarian cancer patients and correlate with poor prognosis and survival [24]. The effector protein of the Hippo pathway, YAP, is a known oncogene in ovarian cancer [25]. Inhibition of YAP causes in vitro and in vivo suppression of platinum therapy resistance [26]. The PI3K/PTEN/AKT pathway is also activated in HGSOC. PI3K inhibition was found to chemosensitise resistant ovarian cancer patients to platinum-based therapy [27]. Patient-derived CD24+ OCSCs showed increased expression of STAT3, and inhibition of the JAK2/STAT3 pathway correlated with better survival [28]. Taxane and JAK2 inhibitor combination therapy was found to cause a decrease in ovarian cancer stemness [29]. The NF-κB pathway has been implicated in the formation of stem cells [30]. Tumorigenic and stemness-initiating properties were verified in a mouse xenograft model-based study which found that stemness was maintained via both canonical and non-canonical cascades of the NF-κB pathway. Inhibition of the pathway restored sensitivity and response to platinum therapy in ovarian cancer cells [31].

### 2.2. Cancer Stem Cell Markers

Cancer stem cells can be identified and confirmed by the presence of specific cell surface and non-surface biomarkers. Several cell surface markers have been associated with OCSCs and are summarised in Table 1.

## 3. The Ovarian Cancer Stem Cell Niche

The intra-tumoral space where stem cells exist and interact with their immediate environment via humoral, neuronal, paracrine, positional and metabolic signals for self-maintenance and overall tumour growth is called the stem cell niche. The cancer stem cell niche interacts with several intra-tumoral processes such as epithelial–mesenchymal transition (EMT), neovascularisation, hypoxic microenvironment and inflammatory networks. The bi-directional communication is biologically dynamic, wherein the cellular processes support the survival, growth and invasive properties of the cells, and the stem cells in turn regulate the cellular processes in the tumour microenvironment for self-benefit.

### 3.1. Epithelial–Mesenchymal Transition

The process by which an epithelial phenotype undergoes transition first by increasing in dimension and subsequently by acquiring a mesenchymal phenotype is called EMT [44]. One of the very first studies identifying stem-cell like subpopulations in the ovarian epithelium by Virant-Klun et al. found strong evidence that the stem cell niche induced EMT [8]. This transition is a dynamic process occurring in conjunction with persisting surrounding epithelial cells, as well as a wide spectrum of stromal cells (fibroblasts, immune cells) and endothelial cells, and enabling invasive and migratory properties within cancer cell populations [45]. Specific transcription factors are associated with the transitional process and can be mainly categorized into three families—TWIST, Snail and ZEB [46]. They suppress epithelial state-inducing genes like E-cadherin and stimulate mesenchymal state-inducing genes like N-cadherin [46]. These transcription factors have also been associated with expression of stemness-enhancing genes [47,48]. In the ovarian cancer stem cell niche, TGF-β signaling plays a significant role in promoting EMT via regulation of tissue transglutaminase 2 (TTGM2) [49]. A dynamic EMT state leads to increased stemness and enables chemoresistance. OCSCs exist in an intermediate epithelial–mesenchymal state, expressing both kinds of markers and equipping them with unique potential for adhesion and migration, respectively [50]. This dials into the plasticity model for cancer stem cells by proving that stemness is a dynamic interconvertible state [51].

### 3.2. Hypoxia

While hypoxia has been implicated as a driver in the maintenance of most cancer stem cell niches, it is of particular interest in ovarian cancer due to the presence of ascites which serve as metastasis hotspots for invasive spheroid formation. Ascites contain half the soluble oxygen as blood [52]. This hypoxic condition stimulates the hypoxia-inducible transcription factor-1 alpha (HIF-1α) to initiate hypoxia-responsive downstream signaling of various target genes which allows cells to adapt to environmental insults. Hypoxia drives stemness [53] and induces chemoresistance potential by maintaining OCSCs in a quiescent state, shielding them from drugs intended to target proliferative cells [54]. HIF-1,2 are involved in stimulating fibroblasts to secrete CXCL12, which is believed to initiate the cancerous phenotype in ovarian cancer. [55] These cells are also able to respond to stress [56], whilst also being invasive and migratory, and can promote increased angiogenic potential [52]. Reactive oxygen species (ROS) are produced by cancer cells and can stimulate oncogenes and facilitate new mutations. A recent study verified that ROS levels were eight times higher in tumours from 34 Stage III/IV HGSOC patients than in non-cancerous ovaries [57]. Elevated level of ROS in cancer stem cells has been found to promote cancer metastasis by inducing EMT via the TGF-β pathway [58,59].

### 3.3. Neovascularisation and Angiogenesis

Hypoxic conditions also induce the expression of vascular endothelial growth factor (VEGF), the most potent pro-angiogenic factor, by various cells (both cancer stem cells and normal cancer cells) in the tumour microenvironment [56]. Specifically, in the ovarian cancer niche, VEGF stimulates the CXCL2 receptor pathway in the endothelial cells, further inducing angiogenesis [60]. Moreover, Alvero et al. showed that OCSCs themselves have the capacity to form new vessels independent of the VEGF pathway [61]. This was further supported in another study demonstrating that OCSCs can self-differentiate into endothelial cells and undergo angiogenesis via activation of NF-κB and JAK2/STAT3 signaling [62]. VEGFA also stimulates upregulation of Bmi1 and loss of miR128-2, which increases stemness [63]. In addition, the vascular niche stimulates the expression of inflammatory cytokines, which further lead to metastatic initiation, and self-renewal and maintenance of the stem cell niche. Hence, the stem cell niche and angiogenic processes trigger and maintain each other in a cyclical manner.

### 3.4. Inflammation

The tumour microenvironment has been linked to chronic inflammation producing cytokines and pro-angiogenic signals which in turn initiate a cascade of immune responses. Immune signaling in the microenvironment, as outlined previously, feeds back into enrichment and maintenance of the stem cell niche, and this aspect will be discussed in further detail.

## 4. Ovarian Cancer Stem Cell Niche and Inflammatory Networks

Although there has been a long apparent association between inflammation and cancer, it was only introduced as one of the ‘Hallmarks of Cancer’ in Hanahan and Weinberg’s second, revised magnum opus [64]. Chronic inflammation has been established as a cause of several cancers [65], and the phenotypes, processes and pathways associated with various immune cells and interactions contribute to the dynamic maintenance of the tumours at a microenvironmental level [66]. These correlations have been verified in vitro, in zebrafish [67] and mouse models [68] as well as in patient prognostic data [69]. Specifically, cancer stem cells can use immune surveillance evasion to enhance their survival and invasive properties. Growing evidence suggests that cancer stem cells are able to not only circumvent key immune checkpoints, but also manipulate inflammatory networks to promote self-sustenance, tumorigenesis and cellular invasion [68]. Ovarian cancer, in particular, is a classic example of a stem cell-driven cancer. It metastasises via a trans-coelomic route spreading to the peritoneal organs in the form of persistent spherical multicellular aggregates. The primary tumour is capable of metastasising very early due to the ability to form spheroids from ascites, which proliferate and persist even in the absence of organ adhesion, and displays key stemness attributes [70]. These cells invade the extracellular matrix where they interact with the cellular microenvironment consisting broadly of immune (cytokines, macrophages, lymphocytes) and non-immune (adipocytes, fibroblasts, endothelial) cell components (Figure 4).

### 4.1. Cytokine Signaling

Not only have cytokines been identified in ovarian cancer patient ascites and cysts [71], they have also been found in the tumour stroma and epithelium [72]. This indicates that active cytokine-mediated signaling is part of the microenvironmental interactions in the ovarian tumour niche. Non-tumoral cells like adipocytes in the omentum and endothelial cells of the vasculature also trigger the release of cytokine signaling. Adipocyte-mediated cytokine signaling induces a change in lipid metabolism and allows cancer cells to use fatty acids as fuel for proliferation [73]. In ovarian cancer, adipocytes express IL-6, increasing the expression of BCLxl that provides the ability to cancer stem cells to become resistant to drug therapy [74]. Endothelial cells on the other hand, enhance inflammation and angiogenetic potential along with cell migration in the tumoral niche via the release of TNF-α, VEGF and interleukins (IL) [75]. IL-17 was one of the first cytokines identified in the ovarian cancer niche which was found to promote self-renewal of OCSCs [76,77]. Upon further investigation, it was found that OCSCs expressed the IL-17 receptor which promotes self-sustenance and growth via the NF-κB and MAPK pathways [76]. The NF-κB pathway has also been implicated via the release of IL-23 [78] and CCL5 [79] by OCSCs, which further enriches the angiogenic potential of tumour cells within the niche.

### 4.2. Tumour-Associated Macrophages (TAMs)

TAMs constitute the highest percentage of immune cells in the tumour niche. JAK2/STAT3 activation within TAMs promotes increased tumorigenicity, chemoresistance and stemness within tumours [80]. Subsequently, anti-tumour CD8+ responses from chemotherapeutic targeting are blocked by the cancer stem cell niche and the polarisation of the TAMs towards an anti-inflammatory M2 phenotype [80]. M2 macrophages in general have been seen to have a notable positive impact on the progression of tumours in different cancers [81]. In particular, among patients with high-grade ovarian cancer, M1 macrophages were significantly associated with better outcomes, while the M2 phenotype was associated with worse outcomes [82]. A co-culture study proved that OCSCs are capable of polarising the macrophage phenotype towards an M2 state via COX-2 overexpression and cytokine production, involving the JAK2/STAT3 pathway [83]. Furthermore, the M2 phenotype stimulates cancer stem cell self-renewal and growth via various signaling pathways e.g., EGF, TGF-β, IL-6 and IL-10, that lead to STAT3 activation [84]. NF-κB signaling pathways are activated, causing subsequent recruitment of M2 macrophages and also contributing to supplementary production of cytokines, and hence feedback into the self-sustaining cycle of the cancer stem cell niche [84]. Furthermore, an immunosuppressive microenvironment may originate as an outcome of the responses of CD4+ Treg T cells that are stimulated by M2 macrophages [85]. Additionally, macrophages make the tumour microenvironment amicable for cancer stem cell seeding as well as migration [86,87].

### 4.3. Tumour-Infiltrating Lymphocytes (TILs)

Tumour-infiltrating lymphocytes (TILs) includes cells such as CD8+ T cells, T regulatory cells and B regulatory cells, and they are recruited to the tumour mass. The presence of these infiltrates in the tumour microenvironment has a varied effect on tumour progression and prognosis, depending on the timeline of tumour growth as well as the subtype of ovarian cancer. In HGSOC, CD8+ T cells were found to correlate with better overall survival [85]. While a significant association was also observed between CD8+ T cells and overall survival in LGSOC, there was no such correlation in endometrioid or clear cell carcinomas [88]. B cells also contribute to tumour regulation both as tumour suppressive immune response cells, and immunosuppressive tumour-promoting cells [89]. In conjunction with T cells, B cells were found to co-localise in the niche, produce markers, and improve overall survival [90]. They also have a counter-regulatory effect on CD8+ T cells [91], contribute to cytokine signaling [92], and hence the overall proportion of these cells correlates with disease progression in a dynamic way.

### 4.4. Natural Killer Cells (NK)

Similar to T cells, NK cells are capable of acquiring memory functional phenotypes once target cells are encountered, thereby bridging the gap between adaptive and innate immune systems [93]. Cancer stem cells can be killed in a major histocompatibility complex (MHC)-unrestricted manner by NK cells [94] via the release of TNF family members [95]. Immunoglobulin Fc, inflammatory cytokines and endogenous ligands activate these NK cells [96]. A range of activating and inhibitory receptors modulate NK cell function. These receptors sense changes, such as loss of MHC in tumour cells, and subsequently allow NK cells to respond accordingly [93]. It was found that OCSCs downregulate NK cell function. Ascites of ovarian cancer patients have been found to have increased levels of NK cells. However, due to the immunosuppressive effects of the ovarian cancer stem cell niche and the dysregulation of natural and cell-mediated cytotoxicity, these cells are functionally impaired [97].

## 5. OCSCs and Immune Surveillance

In the course of disease progression, with the accumulation of mutations, surface markers, known as tumour-associated antigens (TAAs), are released by cancer cells. Hypoxia and tumour necrosis trigger the release of these surface antigens and recruit immune response cells to the tumour microenvironment [98]. Tumour-associated antigens may be differentiation antigens, overexpression antigens or neoantigens [99]. Particularly in ovarian cancer, the Cancer-Testis Antigen (CTA) has been found to be expressed by tumours. CTAs are differentiation tumour antigens expressed in normal testis or healthy placenta tissues, and their dysregulation is associated with abnormal differentiation of OCSCs [100]. Existing research shows that CTA expression in ovarian cancer is caused by DNA hypomethylation [101]. Almost 40% of people suffering from ovarian cancer have the NY-ESO-1 gene of the CTA family expressed in tumours, which has been selectively associated with stem cells [100]. Human epidermal growth factor receptor 2 (HER2) and epidermal growth factor receptor (EGFR) are two examples of overexpressed antigens which have been implicated and negatively associated with prognosis in most epithelial cancers including ovarian cancer [102]. About 30–70% of patients with ovarian cancer demonstrate overexpression of EGFR [103]. These antigens are identified by the immune system to drive pathways that are meant to eradicate these tumoral cells. However, this process is impaired due to the microenvironmental interactions of the OCSC niche.

The cellular immune response has evolved to be effective at recognizing non-self cells and eradicating them from the body. To achieve this in the cancer setting, immune cells need to identify the growing tumour, infiltrate the microenvironment and then proceed to kill tumour cells. However, cancer cells, and especially aggressive, stem cell-laden cancers like ovarian cancers, have the advantage of phenotypic plasticity, genomic instability, fast turnover and hence adaptation rate. Therefore, a number of methods evolve in the tumour niche to evade normal immune surveillance. For example, ovarian cancer cells modify TIL function to create a favourable immunosuppressive microenvironment for the growing tumour. Regulatory T cells (Tregs), which correlate with poor overall prognosis, seem to be extensively recruited by ovarian cancer cells via CCL22 [104] and TGF-β [105] pathways, leading to a suppression of the CD8+ T cell function. The TGF-β pathway has also been implicated in causing dendritic cell dysfunction in the tumour niche via stimulation of the PD-L1 and arginase pathways [106]. Clinically, the interactions between tumour presenting PD-L1 and PD-L2 ligands with their receptors on T cells are some of the most vital pathways manipulated by the ovarian cancer stem cell niche. Upregulation causes a shift in the proportion effector T cells to Tregs, leading to an immunosuppressive microenvironment facilitating cancer proliferation and growth. Increased PD-L1 expression inversely correlated with the number of TILs as well as patient prognosis in ovarian cancer sufferers [107]. In addition to directly decreasing CD8+ T cells, growing tumour microenvironmental interactions led to a decrease in MHC expression, which is identified by the immune surveillance processes initiating a killer immune response [108]. The cytokine and lipid metabolite signaling pathways that are essential in the maintenance of the OCSC niche have been found to be responsible for establishing immunosuppressive environments in the niche, further aiding in the dysfunction of normal immune function. Furthermore, the hypoxic environment present in the stem cell niche coupled with the increased secretion of VEGF and TGF-β drives a dendritic cell phenotype which is tolerogenic. Dendritic cells infiltrate the tumour microenvironment; however, they are immunosuppressed and incompetent [109]. Such an evasion of immune surveillance bears resemblance to interactions between the foetus and the maternal immune system. For example, the CA125 antigen which prevents maternal immune attack on the foetus in the uterus, is overexpressed in ovarian cancer [110].

Metastatic behaviour in ovarian cancer is aggressive but unique, in that ovarian cancer metastasis is early and prevalent, but the spread of the tumour is often restricted to the peritoneal cavity. Having expanded in a diffuse intra-abdominal manner, tumoral invasion remains confined to the peritoneal cavity even after recurrence. Existing research points towards metalloproteinases being present in ascites [111], which degrade the extracellular matrix to make way for the invading tumour. Furthermore, to avoid or combat host immune responses, suppressive mechanisms are enforced by the tumoral niche as explained earlier [112,113]. A scaffold is created using the extracellular matrix and fibroblasts, along with the tumour-infiltrating immune cells, which aids tumour cell expansion. This leads to the growth and nourishment of the tumour due to the creation of an inflammatory milieu [114]. There is also a discernible counterbalance in the weak anti-OCSC immunity generated by T cells. These cells are dysfunctional and contribute to the immunosuppressive environment along with associated cytokine and lipid-based pathways. As a result, although OCSCs are identified by the immune system, they are incapable of being killed, and instead, immune pathways are manipulated in the microenvironment for OCSC growth and migration. Such metastatic behaviour in ovarian cancer indicates that tumour cells cannot survive once they arrive at sites where complete immune responses are viable, having exited the immunosuppressive microenvironment in the peritoneal cavity [115].

## 6. Therapeutic Implications

### 6.1. Targeting Ovarian Cancer Stem Cell-Associated Signaling Pathways

As outlined earlier, recent research has delineated the key signaling pathways associated with ovarian cancer stem cell growth and maintenance. These are some of the most obvious candidates for targeted therapy. Extensive research is underway in the endeavour to translate these into druggable targets. Some Wnt pathway inhibitors such as LGK974 and pri-724 have been found to be effective in various cancers, for example, breast cancer and melanomas [109]. Cyclopamine, an Shh inhibitor, induced a 10-fold decrease in spheroid formation in ovarian cancer cell lines [116]. Vismodegib and sonidegic are SMO inhibitors currently in Phase II trials for ovarian cancer therapy [117]. A gamma-secretase inhibitor of the NOTCH pathway is undergoing Phase 1 clinical trials [118]. Anti-DLL4 antibodies, which prevent ligand binding to the Notch pathway, are being used in advanced ovarian tumours [119]. The therapeutic peptide of FKBPL, an anti-angiogenic protein known to inhibit breast cancer metastasis via a variety of mechanisms, was found to have anti-stemness effects in ovarian cancer cell lines and patient-derived models [120]. Monoclonal antibodies for epithelial cell adhesion molecule (EpCam) are under Phase 3 clinical trial in ovarian cancer immunotherapy [121].

### 6.2. Targeting Ovarian Cancer Stem Cell-Associated Immune Interactions

Immunoregulatory cytokines have been found in ascites of ovarian cancer patients [122]. Administration of recombinant IL-12 has been found to prevent tumour metastasis and has shown promising results in pre-clinical models [123]. Chimeric antigen receptors (CARs) can be recognized by tumour-targeted T cells and trigger specific recognition of antigens on cancer cells to initiate a killer immune response [124]. Additionally, the most recent form of CAR-T therapy utilises IL-12 signaling and is called T cell-redirected universal cytokine killing (TRUCK) [125]. While CAR-T-based therapy has been promising in many cancers, the heterogeneity and plasticity of the stem cell niche in ovarian cancer means that extra efforts need to be made to design CAR-T cells to identify and target OCSCs by optimising for specific markers [126].

An antigen–antibody response is used in monoclonal antibody (mAb) immunotherapy to kill targeted cells by exploiting the immunocompetence of the host [127]. The field of cancer immunotherapy has exploded over the last few years with the development of several mAbs, capable of targeting cell surface proteins on malignant cells [128]. Studies involving ovarian cancer have explored the idea of targeting cancer stem cells using mAbs. This therapy involves activating complement-dependent cytotoxicity and antibody-dependent cell-mediated cytotoxicity, thereby inhibiting memory and effector T cells, priming antigen-presenting cells and receptor-mediated signaling [129]. Existing preclinical research indicates ovarian tumours may be detected at early stages or in treatment using imaging reagents based on mAb CC188 [130]. Furthermore, studies show that the mAb catumaxomab killed CD133+/EpCAM+ cancer stem cells by attaching to tumour cells and T cells in cases of advanced ovarian cancer with malignant ascites [131]. Bevacizumab targets the stem cell-driven VEGF-A pathway to inhibit angiogenesis in the ovarian cancer niche and has been approved by the FDA for use in combinatorial therapy as well as in chemoresistant patients [132].

Another efficient way of stimulating endogenous activation of T cell responses to cancer is by providing the tumour-associated antigen to patients using vaccines. At present, one of the most effective and well-researched methods is dendritic cell-based vaccination. Studies in various tumours have reported the therapeutic efficacy of dendritic cell-based vaccines against cancer stem cells [133]. With respect to ovarian cancer specifically, it has been found that cancer stem cells experienced highly specific anti-tumour T cell responses induced by dendritic cells loaded with NANOG peptides [134]. The therapeutic efficacy of dendritic cell vaccines was verified in mice models for squamous cell cancer, but these results need to be verified in ovarian cancer. Additionally, a study showed blocking the Cxcl2 pathway via oncolytic virotherapy targeted cancer-initiating cells and controlled immunosuppression in ovarian cancer [135]. Another common area for ovarian cancer stem cell-based immunotherapy is the immune checkpoint PD-1 receptor targeted by drugs in the form of monoclonal antibodies, for example, pembrolizumab [132]. Moreover, the anti-VEGF-based therapy approved for anti-angiogenesis-based effects targets the STAT3 pathway and inhibits PD-1 expression [136].

Immunotherapy has been transformational for the treatment of patients with a range of solid tumours. Due to the extensive inflammatory interactions in ovarian cancer, there is vast scope to exploit these in immune-driven therapies. However, due to phenotypic heterogeneity and dynamic pathway regulations, the ovarian cancer stem cell niche is particularly difficult to navigate using immunotherapy. Only two clinical trials targeting the ovarian cancer stem cell niche have been active in the past decade. One study demonstrated that cancer stem cell-primed antibodies were able to target the stem cell niche and establish anti-tumour immunity in patients of ovarian cancer [137]. Furthermore, a more recent Phase II clinical trial study found that metformin causes DNA methylation and prevented stem cell-induced chemoresistance in ovarian cancer patients [138]. Immunotherapeutic targeting of ovarian cancer is being extensively researched and is outside the scope of this review; however, it is important to note that, due to the direct interactions between the stem cell niche and the immune system, emphasis needs to be laid on targeting stem cell-driven pathways for better efficacy and sustained results from such immunotherapies. The limited number of studies directly targeting ovarian cancer stem cell niche brings to light the need for more active clinical trials in this field.

## 7. Conclusions

Cancer stem cells have the capacity to dynamically interact with the tumour microenvironment and host immune surveillance networks. In particular, they are capable of evading key immune processes. Ovarian cancer stem cells, in particular, have been known to establish a characteristic niche, manipulating innate processes in favour of self-renewal as well as metastasis. Due to the myriad of interactions between the cancer stem cell niche and the immune microenvironment, immunotherapy offers a promising area of research. While immunotherapy has been very beneficial against certain types of cancer, in ovarian cancer, it is more difficult to obtain sustained efficacy due to the presence of fast-evolving and plastic stem cell niches. Therefore, more promising strategies are awaited that take advantage of novel specific markers and/or delineation of key pathways and better targeting mechanisms. The dynamic interaction between the immune system and the ovarian cancer stem cell niche means that immune surveillance and evasion can be manipulated intrinsically in the future based on a deeper understanding of bidirectional pathways and by developing sophisticated methods for immunoediting.

## Figures and Tables

**Figure 1 ijms-22-04091-f001:**
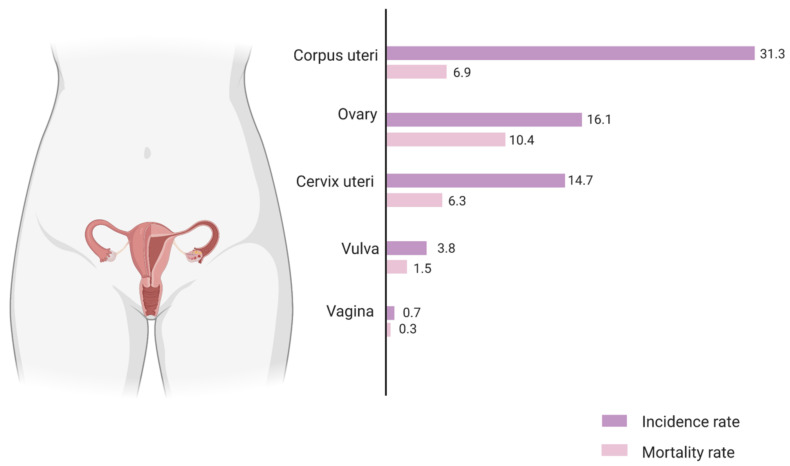
The estimated incidence and mortality rate for gynaecological cancers in European females of all ages, 2020. The values are expressed as age-standardised rate per 100,000 population. The mortality-to-incidence ratio (MIR) for ovarian cancer (0.64) is the highest among all gynaecological cancers and more than twice as high as that for breast cancer (0.25). Source: European Cancer Information System, European Commission.

**Figure 2 ijms-22-04091-f002:**
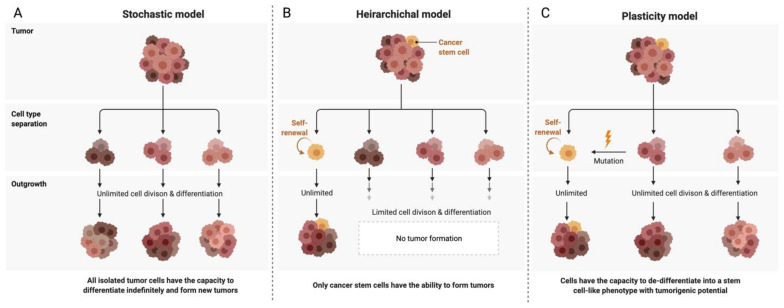
Models of ovarian cancer tumor development and heterogenity. (**A**) The stochastic model—Each cell is considered biologically equivalent (clonal). Heterogeneity is attributed to genetic mutations propagated through time. All cells have tumorigenic capacity. (**B**) The hierarchical model—A single cell undergoes a de-differentiating mutation and forms a distinct subpopulation within the niche having stem cell-like tumorigenic potential and leading to the formation of both intermediate progenitor cells as well as terminally differentiated cells, thus contributing to heterogeneity. (**C**) The plasticity model—Proposes a plastic state of tumorigenic potential in the niche. Differentiated cells can be mutated to re-acquire stem cell-like properties, and the niche contains a dynamic heterogeneous population of differentiated tumour cells as well as stem cells.

**Figure 3 ijms-22-04091-f003:**
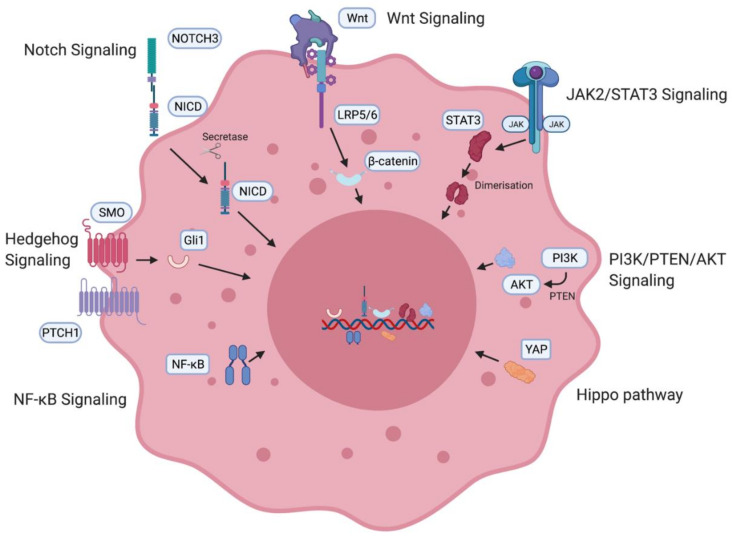
Ovarian cancer stem cell (OCSC)-associated signaling pathways. OCSC signaling pathways involved in the generation and maintenance of OCSCs including the Notch pathway [10,11], Wnt signaling pathway [10,11], JAK2/STAT3 pathway [12,13,14], PI3K/PTEN/AKT pathway [15], Hippo pathway [16], NF-κB [17,18] and the Hedgehog pathway [19]. NICD—intracellular domain of Notch protein; LRP—low-density lipoprotein-related protein; JAK—Janus kinase, STAT—signal transducer and activator of transcription proteins; PI3K—phosphatidylinositol 3-kinase, PTEN—phosphatidylinositol 3,4,5—triphosphate 3-phosphatase, AKT/PKB—protein kinase B; YAP—Yes-associated protein; NF-κB—nuclear factor kappa B.

**Figure 4 ijms-22-04091-f004:**
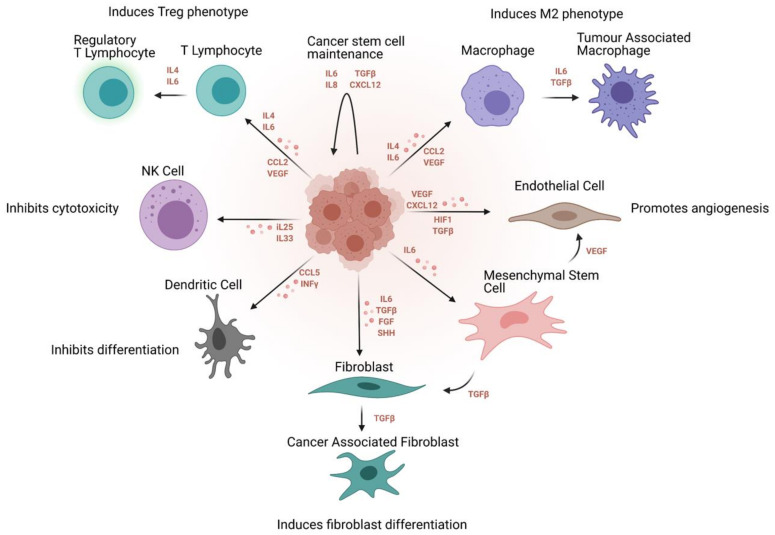
Immunosuppressive effect of the cancer stem cell niche on the tumour immune and non-immune microenvironment. Signaling molecules regulating these processes and the overall effect of the stem cell niche are outlined.

**Table 1 ijms-22-04091-t001:** Markers associated with OCSCs.

Marker	Characteristic	Function in Ovarian Cancer	Evidence
CD133	Transmembrane glycoprotein	Identified by several groups to be expressed in tumour-initiating cells; promotes adhesion to metastatic cancer niche.	Ferrandina et al. [32], Roy et al. [33]
ALDH	Aldehyde dehydrogenase enzyme	Correlates with tumourigenicity and spheroid formation; increased expression significantly associated with poor outcomes in patients with serous ovarian cancer.	Ma et al. [34], Ishiguro et al. [35], Deng et al. [36]
CD44	Transmembrane glycoprotein	Positively associated with ovarian cancer migration and metastatic spread; high expression correlates to recurrence and drug resistance.	Bourguignon et al. [37], Carpenter et al. [38], Sacks et al. [39]
CD24	Glycophosphatidylinositol-anchored membrane glycoprotein	Positive marker; cell lines and tumour samples displayed stemness genes, tumourigenicity, spheroid formation.	Burgos-Ojeda, D. et al. [40], Gao, M.Q. et al. [41]
CD117	Receptor tyrosine kinase	Surface marker binding to stem cell factor; consistently formed tumours in mice models	Mazzoldi et al. [42], Luo et al. [43]

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
