# Peer review of "The Cancer Stem Cell Niche in Ovarian Cancer and Its Impact on Immune Surveillance"

_ijms, 2021, doi:10.3390/ijms22084091_

Round 1

Reviewer 1 Report

The authors propose a review on the cancer stem cell niche in ovarian cancer and its impact on immune surveillance. This is the update of their review published in 2018 in Pharmacology and Therapeutics entitled “Targeting cancer stem cells in the clinic: Current status and perspectives” with a focus on ovarian cancer.

The article is well written and covers all aspects of the proposed subject. However, several parts are not in-depth enough and should be completed (see below).

Major comments:

  1. Figure 3 is not informative enough. It is not possible to summarize signalling pathways by their single name. Even if this diagram becomes complex, it must be completed and allow the interactions between the pathways to be seen. The proteins mentioned in the paragraph must all appear on the diagram.
  2. The pharmacological modulation of stem cell signalling pathways is discussed in section 2. “OCSCs: signalling pathways and markers” and then in the ”therapeutic implications” section. These two parts should be grouped.
  3. section 2. “OCSCs: signalling pathways and markers”: There is no link between the signalling pathways and the proposed markers. The two parts should be separated.
  4. Epithelial-mesenchymal transition paragraph: this paragraph must be completed, other signalling pathways than those of TGF-ß are involved in the process
  5. The roles of hypoxia and HIF-1α on CSCs in relation to all the processes discussed: EMT, angiogenesis, inflammation… are not detailed enough.
  6. Reference to the paper of Ahmed et al. (Semin Cancer Biol. 2018 Dec; 53: 265-281) should be done.

Minor comments:

Several sentences are not very explicit and should be rephrased:

- line 144: This communication…

- line 165: Ascites have half the soluble oxygen as blood

- Line 190: Typo error (linked)

Author Response

Thank you for your revision suggestions. Please find attached the document with point-by-point responses to each comment. A revised version of the manuscript with the inculcated changes is ready to be uploaded. 

Reviewer 2 Report

The manuscript is well organized and well written.

The topic is interesting and exhaustevely described. 

This Reviewer suggests to check typos. Moreover, the Reviewer is wondered about the expression levels of class I MHC molecules, responsible for antigen presentation. Indeed, class I MHC proteins are often down-regulated in tumors and this influence the poor response to CAR-T cell therapy or other immunotherapy approach.

Anyway, in Reviewer's opinion the manuscript is acceptable for publication.

Author Response

Thank you for your review and comments. 

Typos and grammatical mistakes have been identified and corrected. 

The major histocompatibility complex has been touched upon under the subheading OCSC and immune surveillance. In the interest of streamlining the review, it has not been extensively explored. Kindly let me know if you advise more information be added about the same. 

Round 2

Reviewer 1 Report

The manuscript is acceptable for publication.